# Behavioral practices of patients with multiple sclerosis during Covid-19 pandemic

**Hind Alnajashi** *, Razan Jabbad

Neurology Division, Department of Medicine, Faculty of Medicine, King Abdulaziz University, Jeddah, Saudi Arabia

* halnajashi@kau.edu.sa, hindnajashi@gmail.com

**Data Availability Statement:** All relevant data are within the manuscript.

**Funding:** The authors received no specific funding for this work.

## Abstract

Severe acute respiratory syndrome coronavirus 2 (SARS-CoV-2) emerged in Wuhan, China, in late 2019 and Covid-19, a disease caused by SARS-CoV-2 became a pandemic in March 2020. As the pandemic still unfolds, uncertainty circles around the impact of SARS-CoV-2 infection on patients with chronic diseases, including autoimmune diseases such as multiple sclerosis (MS). To diminish the risk of SARS-CoV-2 infection and lessen the impact of the Covid-19 pandemic on the healthcare of MS patients, it is essential to understand knowledge, attitudes, and various behavioral practices related to Covid-19 among MS patients. Therefore, this study aimed to look at the behavioral practices related to Covid-19 among patients with MS. A total of 176 MS patients diagnosed at least one year before the survey were conveniently sampled online in Saudi Arabia and their data collected using a structured interview questionnaire in electronic Google form. We determined the reliability of the questionnaire by measuring its internal consistency in a pilot sample of 30 participants. Overall, more than 80% of participants had good knowledge and attitudes towards Covid-19. However, this did not correlate well with the impact on healthcare (r = 0.06). Our study revealed that 46% of participants were anxious about taking their medication, and 32% of participants missed their hospital appointments. Furthermore, 15% of the participants had a relapse but did not go to the hospital because of the pandemic, 15.9% stopped their DMTs, and 35.2% missed drug infusions or refills. Our study revealed overall good knowledge and attitudes related to Covid-19 among MS patients. However, the healthcare impact was considerable, as 32% of the participants missed their hospital appointments, and another 15% had a relapse. This highlights the significance of the impact of the Covid-19 pandemic on the healthcare of patients with MS. Measures to mitigate the effect of the pandemic on healthcare service delivery to patients with MS, such as telemedicine, should be strongly encouraged.

## Introduction

Severe acute respiratory syndrome coronavirus 2 (SARS-CoV-2) was discovered in Wuhan, China, in late 2019. Covid-19, a disease caused by SARS-CoV-2 quickly spread and was officially declared a pandemic on March 11, 2020 [1]. On March 2, 2020, the Ministry of Health in Saudi Arabia announced the first confirmed case of Covid-19 in the Kingdom of Saudi Arabia. Since

**Competing interests:** The authors have declared that no competing interests exist.

then, the number of cases has steadily increased. As of July 8, 2020, the total number of confirmed cases had reached 220,000, the highest among all the Arabian Gulf States, with 158,000 recoveries and 2059 deaths [2]. As a result of the rapid increase in the number of Covid-19 cases in Saudi Arabia, knowledge, attitudes, and practices related to Covid-19 among citizens will play an essential role in the control of the impact of the Covid-19 pandemic. This is even more essential in populations at high risk of acquiring SARS-CoV-2 infection and to whom the course of Covid-19 would be worrisome, such as patients who require immunosuppressive/immuno-modulatory drugs, including patients known to have Multiple Sclerosis (MS) [3].

MS is an immune-mediated inflammatory demyelinating disease of the central nervous system that affects more than 2 million people around the world and a major cause of severe neurological disability in adult life [4]. Recent studies suggested a moderate-to-high prevalence of MS in the Middle East and in North African countries [5]. Patients with MS require long-term treatment with disease-modifying therapy (DMT) to help control their disease course and decrease the rates of relapse. The type of DMT may differentially affect the risk profile of MS patients with the lowest risk of infections being associated with interferon beta [6, 7]. Some conflicting evidence, however, revealed that most disease-modifying therapies for MS do not particularly target the innate immune system, which confers protection against SARS-CoV-2, and few have any major long-term impact on CD8 T cells to limit protection against Covid-19 [8]. Some other researchers have also shown that immunosuppressive therapy neither appears to have a significant consequence on infection with SARS-CoV-2 nor seems to marshal a severe disease course in many cases [9, 10]. However, it is generally agreed that some drugs, such as interferons, glatiramer acetate, and teriflunomide seem to be safe. On the other hand, B-cell-depleting DMTs, and DMTs which cause lymphopenia such as cladribine, alemtuzumab, and dimethyl fumarate, are cautioned [9, 11]. With the inconclusive evidence of the effect of DMTs on the frequency of SARS-CoV-2 infection and the course of Covid-19, including the serious Covid-19 complications, the decision on whether to discontinue treatment, switching or timing of infusion therapy could be a source of uncertainty to both the patient and the attending physician. Many researchers and organizations have attempted to make recommendations regarding the effect of DMTs on Covid-19, but the evidence is at times conflicting, and a source of confusion [12, 13]. In one survey involving practicing neurologists in North America, a significant proportion of respondents believed that patients with MS are at higher risk of acquiring Covid-19 than the general population. Furthermore, 23% of the respondents were aware that their patients have self-discontinued treatment with disease-modifying therapy due to the fear of SARS-Cov-2 infection, with some doing so against medical advice [14]. With the recent lockdown and postponement of non-essential services in Saudi Arabia, patients with MS may have limited access to routine healthcare services, either because of unavailability or the fear of SARS-CoV-2 infection. Thus, in a challenging time like this, vulnerable groups such as MS patients may be further compromised by the inadequacy of the healthcare system, which put them at considerable risk of relapse and severe neurological symptoms [15].

We conducted this study among individuals known to have MS for at least the past year, living in Saudi Arabia. This study aimed to determine the level of knowledge of the Covid-19 pandemic and attitudes regarding Covid-19 in order to evaluate the extent of the impact Covid-19 has on the healthcare of MS patients and their daily lives. We focused on the effects of the spread of the virus on the patients' hospital visits, medication administration, psychosocial implications, risk of acquiring the SARS-Cov-2 infection, and finally, the appropriate preventive behaviors.

## Materials and methods

### Study design

We conducted a descriptive cross-sectional survey to evaluate knowledge, attitudes, and practices related to the Covid-19 pandemic in MS patients living in Saudi Arabia. This survey took place in June 2020, during the peak of the Covid-19 pandemic in the Kingdom of Saudi Arabia. Our study was conducted during the curfew in Saudi Arabia. We included MS patients who, during our study period, had been diagnosed with MS for at least one year.

### Study participants and recruitment

Since this was a descriptive cross-sectional study, the sample size was determined by a convenient sampling technique, based on the availability of MS patients. A total of 176 patients agreed to participate. An electronic Google form was sent to the participants. The questionnaire was sent through WhatsApp and posted on the Twitter account of the principal investigator in addition to two other MS neurologists practicing in Saudi Arabia. An informed written consent was included at the beginning of the survey. To ensure patients privacy, no personal identifications were included in the form and responses were encrypted to protect the data.

### Data collection and statistical analysis

Data were collected using a structured interview questionnaire. The questionnaire was designed by the authors to cover issues related to the Covid-19 pandemic on people living with MS. The survey questions were based on emerging COVID-19 reports, guidelines, clinical experience, and incorporated patients' inquiries during the pandemic. Eligible participants had to answer questions about their demographic details, disease duration, and type of medications used. This was followed by three sections to assess knowledge, attitude, and practices related to the COVID-19 pandemic. Knowledge was assessed by three questions based on Covid-19 symptomatology and preventive measures. Score 3,2,1 for Yes, To some extent and No respectively, maximum score is 9 and minimal score is 3. Score above 7 were considered as good knowledge and score above 3 as average and 3 as poor knowledge. Attitudes were assessed by asking participants about hand washing, social distancing, and quarantine rules. Answers graded 3,2,1, for always, sometimes and never respectively with score above 7 were considered as good attitude. Practices related to Covid-19 were assessed by asking seven questions about the impact of Covid-19 on practices related to the clinical management of MS. The responses were scaled Yes, To some extent and No with 3, 2,1 respectively and reversing the score for negative item. The questionnaire's reliability was assessed by measuring its internal consistency, which was done by pre-testing the survey on a small sample of the study population (30 participants). Internal consistency reliability was measured using IBM SPSS 23.0 software to determine Cronbach's Alpha value for each part, and the questionnaire was reliable if Cronbach's Alpha value was found to be $\geq 0.7$ in all of the three sections of the questionnaire. Results showed that all the three sections showed internal consistency with Cronbach's Alpha value of $\geq 0.7$ after some adjustments in the questions were made.

The responses to the questionnaires were extracted from Google Forms and exported to Microsoft Excel for coding. The coded data were exported to SPSS version 23 for analysis. We calculated descriptive statistics (means with SDs and percentage frequencies) to characterize the distribution of the study results. We received a total of 192 responses. After reviewing 16 responses were excluded because of duplication or missing responses.

### Ethical approval

The study received ethical approval from the King Abdul-Aziz University Hospital's Institutional Review Board.

## Results

(Table 1: Demographic and other various characteristics of participants) summarizes the demographic characteristics of the participants. A total of 176 patients with MS responded to the questionnaire. Of the 176, 122 (69%) participants were female, with a mean age of 32±9.2 years. Out of 176 participants, 168 (95.5%) had relapsing-remitting multiple sclerosis (RRMS), and a small percentage had primary progressive multiple sclerosis (PPMS) and secondary progressive multiple sclerosis (SPMS). The disease duration ranged from 1 to 25 years, with a mean duration of 5.5±5.1 years. Most of the patients were on disease-modifying therapy (92%). The most frequently prescribed medications were injectable interferons (26%), followed by Ocrelizumab (20.5%), while the least-used therapy was Almetuzemab (1.1%) (Fig 1). A majority of the participants (75%) were dependent on governmental health care systems for medication and follow-up, 20% had private insurance, and only 4.5% paid for their health care out of pocket.

**Table 1. Demographic and other various characteristics of participants.**

| Variables | Percentages |
|---|---|
| Age (Years) | Mean ± SD |
| | 32.51±9 |
| Sex | |
| Male | 54 (30.7%) |
| Female | 122 (69.3%) |
| Education | |
| High School | 30 (17%) |
| Bachelor | 114 (64.8%) |
| Post-graduate | 32 (18.2%) |
| Employment | |
| Employed | 104 (59.1%) |
| Unemployed | 72 (40.9%) |
| Type of MS | |
| RRMS | 168 (95.5%) |
| PPMS | 4 (2.3%) |
| SPMS | 4 (2.3%) |
| Duration of the disease (Years) | (Mean ± SD) |
| | 5.5 ± 5 |
| Medication provider | |
| Governmental hospital | 132 (75.0%) |
| Private insurance | 36 (20.5%) |
| Own expense | 8 (4.5%) |

MS; Multiple Sclerosis, RRRM; Relapsing-Remitting Multiple Sclerosis, PPMS; Primary-Progressive Multiple Sclerosis, SPMS; Secondary-Progressive Multiple Sclerosis, PRMS; Progressive-Relapsing Multiple Sclerosis.

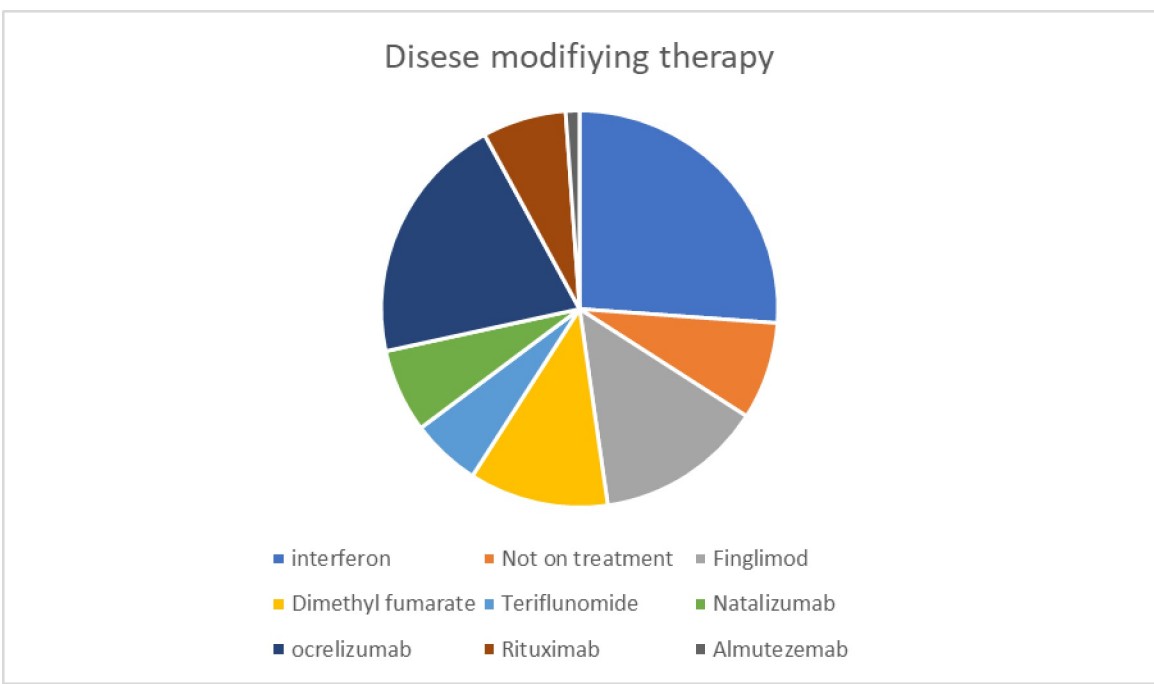

**Fig 1. Types of disease-modifying therapy used by participants.**

### COVID-19 knowledge among MS patients

The mean knowledge score was 8.6, and 92.1% scored above 8, which is considered sufficient knowledge. All the patients were aware of the pandemic, and 168 (95%) of the participants knew that there was a pre-symptomatic period. Regarding the knowledge about precautions required to prevent the spread of Covid-19, 158 (89.8%) participants understood the precautions. Some participants, 18 (10.2%), knew the precautions to some extent, and none of the participants did not understand preventive measures at all.

### Attitudes related to Covid-19 among MS patients

Our participants had a good attitude overall. The majority of participants (n = 146, 83%) knew and followed the ministry of health home quarantine instructions. Of all participants, 158 (89.8%) mentioned that they regularly washed their hands. Most participants, that is 148 (84.1%), followed social distancing rules regularly. Concerning the quarantine guidelines, 144 (81.8%) participants responded that they always stayed at home.

### Healthcare practices during the Covid-19 pandemic among MS patients

In assessing the impact of the Covid-19 pandemic on the healthcare of patients with MS, 94 (53.4%) participants stated that they do not have any fear or concern with continuing their disease-modifying therapy and, 48 (27.3%) had some fear, while only 34 (19.3%) were anxious about taking their medications. Of all participants, only 28 (15.9%) completely stopped taking their medications because of the fear of Covid-19 and 10 (5.7)% stopped taking their medication to some extents which mean poor compliance, but only 6 (3.4%) participants asked their doctors to discontinue or change their disease-modifying therapy and 2 (1.1%) stated that to some extent they raised concerns to their to doctors. Regarding the effect of

the curfew hours on accessibility to health care facilities, 76 (43.2%) participants had to miss or cancel their appointments, and 56 (31.8%) participants missed their appointments because they wanted to avoid any possible source of SARS-Cov-2 infection. It is also worth noting that 26 (15%) participants mentioned they had a relapse and did not go to the hospital because of the Covid-19 pandemic. Another 62 (35%) participants mentioned that their infusion therapy date or medication refill was delayed because of the Covid-19 pandemic. (Table 2 summarizes the responses of the participants).

**Table 2. Participants' responses.**

| Part 1: Participants' knowledge related to Covid-19. | |
|---|---|
| Did you know that it may take up to 10 days to develop symptoms? | Yes 168 (95.5%) |
| | To some extent 8 (4.5%) |
| Do you know the precautions needed to avoid infection with Covid-19? | Yes 158 (89.8%) |
| | To some extent 18 (10.2%) |
| Do you follow MOH quarantine instructions? | Yes 146 (83%) |
| | To some extent 30 (17%) |
| **Part 2: Participants' attitudes related to Covid-19.** | |
| Do you wash your hands regularly? | Always 158 (89.8%) |
| | Sometimes 18 (10.2%) |
| Do you follow social distancing rules? | Always 148 (84.1%) |
| | Sometimes 28 (15.9%) |
| Do you stay home all the time? | Always 144 (81.8%) |
| | Sometimes 32 (18.2%) |
| **Part 3: Practices related to the impact of Covid-19.** | |
| Are you scared to take your medication? | Yes 34 (19.3%) |
| | No 94 (53.4%) |
| | To some extent 48 (27.3%) |
| Did you stop taking your medication because of the pandemic? | Yes 28 (15.9%) |
| | No 138 (78.4%) |
| | To some extent 10 (5.7%) |
| Did you ask your doctor to change your disease-modifying therapy? | Yes 6 (3.4%) |
| | No 168 (95.5%) |
| | To some extent 2 (1.1%) |
| Did you have to change your appointment or miss it because of the curfew? | Yes 76 (43.2%) |
| | No 78 (44.3%) |
| | To some extent 22 (12.5%) |
| Did you have to change your appointment or miss it to avoid getting infected? | Yes 56 (31.8%) |
| | No 96 (54.5%) |
| | To some extent 24 (13.6%) |
| Did you avoid going to the hospital because of the pandemic despite having a relapse? | Yes 26 (14.8%) |
| | No 136 (77.3%) |
| | To some extent 12 (6.8%) |
| Did the pandemic lead to any change in your OPD, drug infusion, or medication refill? | Yes 62 (35.2%) |
| | No 82 (46.6%) |
| | To some extent 32 (18.2%) |

## Association between knowledge, attitudes, and healthcare practices among MS patients

We examined the correlation between knowledge, attitude, and healthcare practices. The mean knowledge score was 8.6±0.6, the mean attitude score was 8.5±0.7, and the mean score for the impact of the pandemic on healthcare was 11.1±3.6. Pearson's correlation coefficient revealed a moderate positive correlation between knowledge and attitude (r = 0.6). However, the correlation between knowledge and healthcare practices was weak (r = 0.06).

## Discussion

Covid-19, as it ravages across the globe, has carried, alongside its wave of devastation, demand for new knowledge, and a dire need of novel public health policy measures. Despite the rapid accumulation of scientific knowledge regarding the microbiology of SARS-CoV-2, and also the pathogenesis and the clinical course of Covid-19, public health measures in many countries have crumbled. As policymakers around the world struggle to combat the rapid spread of the Covid-19 pandemic, they find themselves in unchartered territory, not only because of the laxity of the healthcare system, but also the influence of public tension accompanying a novel life-threatening disease, which frequently leads to wrong decision-making. The Covid-19 pandemic has blatantly exposed the vulnerabilities of the global healthcare system in terms of fragmented responses, unpreparedness, and poor sustainability [16]. The uncertainty around public health policies has also emerged as policymakers struggle to concurrently constrain the disease spread and ameliorate the consequential economic meltdown associated with some public health policies. Countries have thus applied different practices and policies to stifle the spread of the disease in its infancy and mitigate the imminent economic consequences. Social distancing, quarantine, and isolation, when stringent, have proved effective in containing the disease [17]. However, far from their economic impacts, these measures have surfaced various challenges in public health, which are partly fueled by the growing public tension due to misinformation, myths, and insufficiencies in healthcare [18]. Surveys conducted in different countries reported moderate to severe anxiety attributed to the pandemic in 23%, 19%, and 12% of the respondents from Saudi Arabia, Iran, and Spain, respectively [19–21]. Anxiety levels could be higher in patients with chronic diseases as mortality, in general, is higher in the population of people with chronic diseases [22]. Other studies of the general population revealed lower psychological well-being and higher scores of anxiety and depression compared to before Covid-19 pandemic [23]. The routine healthcare of patients with chronic diseases has also been adversely affected, especially when resource-deficient countries reallocate their meager resources to the Covid-19 pandemic, thereby disrupting the routine continuum of care [24]. In patients with MS, misconceptions about the risk of infection with a highly-contagious virus such as SARS-CoV-2 can lead to wrong decisions such as avoiding hospital visits when it is necessary or discounting medications. In dealing with these dynamic challenges, specific local measures that address the needs of a particular population are required. In instituting these measures, a clear exposure of the knowledge gap, attitudes related to Covid-19, and various behavioral practices have to be considered. In this study, we report these key issues with respect to MS patients and suggest possible interventions.

We interviewed a total of 176 respondents who completed the survey, out of whom 95.5% understood the symptomatology of Covid-19, 90% were following precautions instituted by the ministry of health, and 83% were obeying quarantine regulations. These findings reflect a high level of knowledge of Covid-19 symptoms and practice preventive measures among MS patients. These findings are also similar to those of other studies that were done in other countries [25–27]. However, the high level of knowledge in our study should be interpreted with

caution, as all of our participants were recruited online through WhatsApp and Twitter platforms, which were somewhat undoubtedly among the major sources of information, especially during the curfew [28–30]. Most of our participants (64%) were Bachelor's degree holders who may be reasonably assumed to have above-average knowledge. In this case, the level of knowledge in an all-inclusive MS population might have been slightly lower. Of all participants, 90% washed their hands regularly, 84% followed social distancing precautions, and 82% always stayed at home during the curfew. The high level of knowledge was, therefore, reflected in their attitudes towards reducing the risk of SARS-CoV-2 infection, as depicted by a moderately-positive Pearson's correlation. Most participants did not stop taking their DMTs (78.4%), even though 27.3% of them experienced some anxiety towards SARS-CoV-2 infection. Although our participants had an appreciable level of knowledge and an overall good attitude towards Covid-19, there was still some level of anxiety among them. Fear and anxiety from taking disease-modifying therapy were present in 46% of the participants in this survey. Around 15% of the participants avoided going to the hospital despite experiencing relapse symptoms, which might have been accounted for by missed hospital appointments and drug infusions, or self-centered decisions to quit DMTs. Acute relapses, despite posing various challenges in the management of MS, can affect the progression of the disease and frequently leave patients with clinical or subclinical sequelae [31]. Other researchers elsewhere have also reported high levels of anxiety in some high-risk populations in even greater figures [32]. Although most MS patients did not stop DMTs, the proportion which stopped them is likely to be accommodated in the 95.5% of participants who did not raise the concern to their physicians, which suggests that their decision was likely self-centered. Of all, 54.5% of participants opted to change their appointments to avoid SARS-CoV-2 infection, although it was not clear if they missed important follow-ups that could negatively impact their health. Likewise, almost half (43.2%) of all participants changed their hospital appointments, this time because of the curfew. This highlights our concern that public health policies during the Covid-19 pandemic might have led to some other detrimental consequences in marginalized groups. Around 47% of participants missed drug infusion or medication refill schedules because of the curfew, patient's choice, or irregularities in the healthcare system during the Covid-19 pandemic. These results emphasize the fact that the Covid-19 pandemic impacted patients with chronic diseases such as MS in several aspects, raising the need for developing feasible solutions to these groups and reforming the healthcare system in preparation for future pandemics. Our results also raise the need to promote innovative technologies such as telemedicine in delivering healthcare for such populations, especially in an era of a pandemic such as Covid-19, who are not only at high risk but might have other constraints such as neurological deficits, which may further hinder their mobility and access to healthcare [33–36]. Several researchers have spotlighted the potential role of telemedicine in providing healthcare services, especially to marginalized groups, including MS patients [37, 38] who are most likely to be affected by the Covid-19 pandemic, which should permanently be embedded in the traditional healthcare for future pandemics [39, 40]. More than half (52.3%) of the participants received their information from their physicians, which emphasizes the fact that physicians are the most trusted source of information for patients with chronic diseases such as MS. This also stresses on the utility of telemedicine in ensuring real-time communication of health information between patients and their doctors during pandemics. In a chronic autoimmune disease that needs long-term treatment like MS, the optimal timing and choice of disease-modifying therapy could be challenging, especially so during the Covid-19 pandemic. Furthermore, there are no clear guidelines, and it is left to the treating neurologist based on their own experience and anecdotal clinical reports. In a survey including physicians treating patients with MS, one of the neurologists replied to an open question: "This is the most difficult time I have seen in my

19 years of neurology" [14]. Fortunately, data are accumulating rapidly, and we are learning more about Covid-19. In the beginning of the pandemic, some experts recommended delaying treatment with some highly efficacious therapy such as Cladribine, Ocrelizumab, and Alemtuzumab [41]. However, recent data suggested that the outcome of patients with MS who get SARS-CoV-2 infection is not different from that of the general population [42]. Physicians should be vigilant to updates and recommendations regarding MS to guide their patients during the Covid-19 pandemic.

## Conclusion

Our study explored the knowledge, attitudes, and various aspects of the impact of Covid-19 in the healthcare of people with MS. We found an overall good understanding of the Covid-19 pandemic, with the vast majority of patients following health precautions to mitigate the risk of SARS-CoV-2 infection. We also found that the Covid-19 pandemic has a significant impact on the healthcare of patients with MS. Some patients opted to stop their DMTs without medical advice; others missed hospital appointments, drug infusion or refills, and experienced relapse symptoms. Although it is still inconclusive whether Covid-19 has a worse clinical course in MS patients or not, the risk of SARS-CoV-2 infection is generally high, and the impact of Covid-19 due to risky behavioral practices because of the fear of SARS-CoV-2 infection may be far-reaching. In realizing that these consequences were partly attributed to the frailty of the healthcare system, we have suggested some solutions to them. As it is still unknown when the Covid-19 pandemic will come to an end, telemedicine adoption, probable home infusion options, and giving more time to address patient queries is the best way to minimize the impact of the pandemic on this vulnerable group of the society. Altogether, our results provide a framework for public policy actions in resolving problems associated with the impact of Covid-19 in MS patients residing in Saudi Arabia.

## Acknowledgments

We would like to thank Dr. Rumaiza Alyafeai and Dr. Foziah Alshamrani for their contribution to disseminating the survey questionnaire by posting it on their Twitter accounts.

## Author Contributions

**Conceptualization:** Hind Alnajashi, Razan Jabbad.

**Formal analysis:** Hind Alnajashi.

**Methodology:** Hind Alnajashi.

**Supervision:** Hind Alnajashi.

**Writing – Original Draft:** Razan Jabbad.

**Writing – review & editing:** Hind Alnajashi.

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
