## [Decision Letter · Decision Letter 0]

21 Aug 2020

PONE-D-20-22758

"Behavioral practices of  patients with Multiple Sclerosis during Covid-19 pandemic."

PLOS ONE

Thank you for submitting your manuscript to PLOS ONE. After careful consideration, we feel that it has merit but does not fully meet PLOS ONE’s publication criteria as it currently stands. Therefore, we invite you to submit a revised version of the manuscript that addresses the points raised during the review process.

We look forward to receiving your revised manuscript.

Kind regards,

Luigi Lavorgna

Academic Editor

PLOS ONE

Journal Requirements:

2) Please amend either the title on the online submission form (via Edit Submission) or the title in the manuscript so that they are identical.

3) Please remove your figures from within your manuscript file, leaving only the individual TIFF/EPS image files, uploaded separately.  These will be automatically included in the reviewers’ PDF.

Reviewers' comments:

Reviewer's Responses to Questions

**Comments to the Author**

1. Is the manuscript technically sound, and do the data support the conclusions?

Reviewer #1: Yes

2. Has the statistical analysis been performed appropriately and rigorously? 

Reviewer #1: Yes

3. Have the authors made all data underlying the findings in their manuscript fully available?

Reviewer #1: Yes

4. Is the manuscript presented in an intelligible fashion and written in standard English?

Reviewer #1: Yes

5. Review Comments to the Author

Reviewer #1: Revisions to PONE-D-20-22758

In the research article "Behavioral practices of patients with Multiple Sclerosis during Covid-19 pandemic." the Authors aimed to investigate knowledge, attitudes and behavioral practices related to Covid-19 among patients with MS . The main results revealed overall good knowledge and attitudes related to Covid-19 among MS patients. However, the healthcare impact was considerable, as 32% of the participants missed their hospital appointments, and another 15% had a relapse and did not refer the hospital because of the pandemic.

The Authors conclude that Covid-19 pandemic impacted patients with MS in several aspects, raising the need for developing feasible solutions and reforming the healthcare by promoting innovative technologies such as telemedicine in delivering healthcare for such populations who may be not only at high risk but might have other constraints such as neurological deficits, which may further hinder their mobility and access to healthcare.

The manuscript is well written and easy to read; methods are sound and conclusions supported by the results. However, some minor revisions are needed.

Abstract

Line 28: in the sentence “Therefore, this study aimed to behavioral practices ….” It seems that something is missing

Line 37: in the sentence “Furthermore, 15% of the participants had a relapse….” it should be specified that 15% had a relapse and did not go to the hospital because of the Covid-19 pandemic, otherwise it seems that the COVID-19 was responsible for the relapse

Introduction

Line 67: in addition to reference # 7, refer to the article by Ghajarzadeh M. “Are patients with multiple sclerosis (MS) at higher risk of COVID-19 infection?” Neurol Sci. 2020

Line 75: in addition to reference #9, refer to the article by Zheng C et al “Multiple Sclerosis Disease-Modifying Therapy and the COVID-19 Pandemic: Implications on the Risk of Infection and Future Vaccination. CNS Drugs. 2020

Line 98: Please delete the sentence “We also looked at the impact of the Covid-19 pandemic on the healthcare services for patients with MS.” since the methods were not designed to directly assess the impact on healthcare services for patients with MS

Line 99: Please move the sentence “This survey took place in June 2020, during the peak of the Covid-

19 pandemic in the Kingdom of Saudi Arabia” to the methods section

Study participants and recruitment

It is necessary to describe how privacy of the patients was protected

Line 114: the sentence “who were exclusively patients diagnosed with MS for one year or more and were living in Saudi Arabia” is a repetition and should be deleted

Data collection and statistical analysis

A translated version of the questionnaire should be available in this section, also as supplementary material

Results

If there were missing responses it should be specified at the beginning of the methods and numbers about the missing data should be given

Healthcare practices during the Covid-19 pandemic among MS patients

In this section could you provide answers stratified for disease-modifying therapy? It would be interesting to know whether patients on injectables drugs were less anxious than those on depletive therapy

Discussion

Percentages reported in the discussion are reversed or different with respect to those reported in the results section: it would be easier for the reader to follow the discussion with the same values reported in the results

Lines 258-260: the sentence “Around 21.6% of the participants avoided going to the hospital despite experiencing relapse symptoms” is not in agreement with results showing a 15% of patients experiencing a relapse (as reported in lines 187-188)

Lines 254-255: again in this sentence “Most participants did not stop taking their DMTs (78.4%), …..” the 78.8% is not in line with the results reported in line 184 “Of all participants, only 28 (15.9 %) stopped”; please explain this discrepancy.

Line 266-267: again the sentence ”…. 95.5% of participants who did not raise the concern to their physicians” the 95.5% is not in line with the results reported in line 182-183 (“only 6 (3.4%) participants asked their doctors to discontinue or change their disease-modifying therapy”)

Line 282: in addition to reference 32, please refer to please refer to these three articles:

1) Assessing disability and relapses in multiple sclerosis on tele-neurology by Moccia et al. Neurol Sci. 2020

2) Feasibility of Real Time Internet-Based Teleconsultation in Patients With Multiple Sclerosis: Interventional Pilot Study by D'Haeseleer et al. J Med Internet Res. 2020 Aug

3) Digital triage for people with multiple sclerosis in the age of COVID-19 pandemic by Bonavita et al Neurol Sci 2020

Line 284: as a reference to support the sentence “Several researchers have spotlighted the potential role of telemedicine in providing healthcare services, especially to marginalized groups, including MS patients” add:

1) E-Health and multiple sclerosis: An update. Lavorgna et al. Mult Scler. 2018

2) E-health and multiple sclerosis. Matthews et al Curr Opin Neurol. 2020

6. PLOS authors have the option to publish the peer review history of their article (what does this mean?). If published, this will include your full peer review and any attached files.

Reviewer #1: No

---

## [Author Response · Author response to Decision Letter 0]

7 Oct 2020

Abstract

Line 28: in the sentence “Therefore, this study aimed to behavioral practices ….” It seems that something is missing

 Therefore, this study aimed to look at the behavioral practices related to Covid-19 among patients with MS .

The missing word “ look at” has been added. 

Line 37: in the sentence “Furthermore, 15% of the participants had a relapse….” it should be specified that 15% had a relapse and did not go to the hospital because of the Covid-19 pandemic, otherwise it seems that the COVID-19 was responsible for the relapse 

We added this information to clarify the sentence.“and did not go to the hospital because of the pandemic”

Introduction

Line 67: in addition to reference # 7, refer to the article by Ghajarzadeh M. “Are patients with multiple sclerosis (MS) at higher risk of COVID-19 infection?” Neurol Sci. 2020

Reference added. 

Line 75: in addition to reference #9, refer to the article by Zheng C et al “Multiple Sclerosis Disease-Modifying Therapy and the COVID-19 Pandemic: Implications on the Risk of Infection and Future Vaccination. CNS Drugs. 2020

Reference added. 

Line 98: Please delete the sentence “We also looked at the impact of the Covid-19 pandemic on the healthcare services for patients with MS.” since the methods were not designed to directly assess the impact on healthcare services for patients with MS

Agree with you, deleted . 

Line 99: Please move the sentence “This survey took place in June 2020, during the peak of the Covid-19 pandemic in the Kingdom of Saudi Arabia” to the methods section

Moved to method section 109,110 .

Study participants and recruitment

It is necessary to describe how privacy of the patients was protected 

To ensure patients privacy ,no personal identifications were included in the form and responses were encrypted to protect the data. ( We added this sentence to the method section) 

Line 114: the sentence “who were exclusively patients diagnosed with MS for one year or more and were living in Saudi Arabia” is a repetition and should be deleted

This sentence has been deleted. 

Data collection and statistical analysis

A translated version of the questionnaire should be available in this section, also as supplementary material. 

It is added as a supplementary material ,and most of the content of the questionnaire are available on Table 2 .

Results

If there were missing responses it should be specified at the beginning of the methods and numbers about the missing data should be given. 

We received a total of 192 responses. After reviewing 16 responses were excluded because of duplication or missing responses. I added this information to the end of the method section. 

Healthcare practices during the Covid-19 pandemic among MS patients

In this section could you provide answers stratified for disease-modifying therapy? It would be interesting to know whether patients on injectables drugs were less anxious than those on depletive therapy. 

We tried to stratify the patients into low risk medication (injectable interferons ) and higher risk medication (all the other DMTs ) Interestingly there were no significant difference between the two groups . The sample sized is relatively small to yield a meaningful result from stratifying to injectable , oral , infusion or comparing the response to each drug alone. 

Discussion

Percentages reported in the discussion are reversed or different with respect to those reported in the results section: it would be easier for the reader to follow the discussion with the same values reported in the results

Lines 258-260: the sentence “Around 21.6% of the participants avoided going to the hospital despite experiencing relapse symptoms” is not in agreement with results showing a 15% of patients experiencing a relapse (as reported in lines 187-188)

Corrected to 15 % of participants avoided going to the hospital despite having a relapse . 

Lines 254-255: again in this sentence “Most participants did not stop taking their DMTs (78.4%), …..” the 78.8% is not in line with the results reported in line 184 “Of all participants, only 28 (15.9 %) stopped”; please explain this discrepancy.

 The questions Did you stop taking your medication because of the pandemic ? could be answered as Yes ,to some extent or No. 

78.% did not stop taking their medication , 15.9% answered Yes they stopped the medication and 5.7% stopped their medication to some extent . 

We added this to the results section: (only 28 (15.9 %) completely stopped taking their medications because of the fear of Covid-19 and 10 (5.7)% stopped taking their medication to some extents which means poor compliance.) 

Line 266-267: again the sentence ”…. 95.5% of participants who did not raise the concern to their physicians” the 95.5% is not in line with the results reported in line 182-183 (“only 6 (3.4%) participants asked their doctors to discontinue or change their disease-modifying therapy”) 

This was the response to this question : 

Yes 6 (3.4 %)

No 168 (95.5 %)

To some extent 2 (1.1 %)

I changed the sentence in results section to only 6 (3.4%) participants asked their doctors to discontinue or change their disease-modifying therapy and 2 (1.1%) stated that to some extent they raised concerns to their to doctors.

Line 282: in addition to reference 32, please refer to please refer to these three articles:

1) Assessing disability and relapses in multiple sclerosis on tele-neurology by Moccia et al. Neurol Sci. 2020

2) Feasibility of Real Time Internet-Based Teleconsultation in Patients With Multiple Sclerosis: Interventional Pilot Study by D'Haeseleer et al. J Med Internet Res. 2020 Aug

3) Digital triage for people with multiple sclerosis in the age of COVID-19 pandemic by Bonavita et al Neurol Sci 2020

Line 284: as a reference to support the sentence “Several researchers have spotlighted the potential role of telemedicine in providing healthcare services, especially to marginalized groups, including MS patients” add:

1) E-Health and multiple sclerosis: An update. Lavorgna et al. Mult Scler. 2018

2) E-health and multiple sclerosis. Matthews et al Curr Opin Neurol. 2020 

All references are added .

Thanks for your time to review and valuable comments.

---

## [Editor Report · Decision Letter 1]

9 Oct 2020

"Behavioral practices of  patients with Multiple Sclerosis during Covid-19 pandemic."

PONE-D-20-22758R1

Dear Dr. Hind Alnajashi

We’re pleased to inform you that your manuscript has been judged scientifically suitable for publication and will be formally accepted for publication once it meets all outstanding technical requirements.

Kind regards,

Luigi Lavorgna

Academic Editor

PLOS ONE

---

## [Editor Report · Acceptance letter]

13 Oct 2020

PONE-D-20-22758R1 

"Behavioral practices of patients with Multiple Sclerosis during Covid-19 pandemic "  

Dear Dr. Alnajashi:

I'm pleased to inform you that your manuscript has been deemed suitable for publication in PLOS ONE. Congratulations! Your manuscript is now with our production department. 

Kind regards, 

on behalf of

Dr. Luigi Lavorgna 

Academic Editor

PLOS ONE